# *Staphylococcus aureus agr*-type vs genetic background: molecular signatures determining differential metabolism and virulence potential

**Mariane Pivard**[ID]ʘ, **Julian Bär**[ID]ʘ, **Tomas Demeter**, **Srikanth Mairpady Shambat**, **Annelies S. Zinkernagel***

Department of Infectious Diseases and Hospital Epidemiology, University Hospital Zurich, University of Zurich, Zurich, Switzerland

ʘ These authors contributed equally to this work.
* annelies.zinkernagel@usz.ch

## Abstract

In *Staphylococcus aureus*, the quorum-sensing accessory gene regulator (*agr*) system is the major virulence regulator. The four *agr*-types (I-IV) have been associated with distinct infection outcomes, but their direct contribution to virulence regulation and metabolism has remained unresolved due to tight linkage between *agr*-type and genetic background. To disentangle *agr*-type-specific effects, we used congenic Newman strains in which the native *agr*-locus has been replaced with each of the four *agr*-types, alongside a Δ*agr* mutant. We performed RNA-sequencing during early exponential (1h30), late exponential (6h), and stationary (12h) growth phases. Despite similar growth kinetics, *agr*-types displayed distinct activation profiles based on *agrA* and RNAIII expressions. *Agr*-I and *agr*-IV showed early, strong expressions, *agr*-II displayed intermediate expressions and *agr*-III initiated weak expressions only in stationary phase. This *agr*-type-dependent activation timing was the dominant driver of global transcriptional changes. Early activation in *agr*-I and *agr*-IV induced robust expression of phenol-soluble modulins, capsule biosynthesis genes, and pore-forming toxins, whereas *agr*-II and *agr*-III expressed delayed or alternative virulence pathways, including upregulation of superantigen-like genes. Among all types, *agr*-IV exhibited the broadest transcriptional response, encompassing both virulence and metabolic pathways, including differential regulation of nucleotide and fructose metabolisms. Pairwise differential expression, over-representation analysis, and gene-set enrichment consistently revealed *agr*-type-specific virulence and metabolic programs. *Agr*-III, which activated latest and weakest, showed limited transcriptional change until stationary phase, whereas *agr*-I, *agr*-II, and *agr*-IV displayed progressively broader virulence and metabolic remodeling. Together, these findings demonstrate that *agr*-type determines virulence and metabolic gene expression profiles primarily by dictating the timing and magnitude of *agr*-activation, even within an identical genetic background and growth environment. This work provides a systematic

**Data availability statement:** The raw reads for this study have been deposited in the European Nucleotide Archive (ENA) at EMBL-EBI under accession number PRJEB107700 (https://www.ebi.ac.uk/ena/browser/view/PRJEB107700).

**Funding:** A.S.Z. was supported by the Swiss National Science Foundation (SNSF) project grant 310030_204343 and M.P. by the Promedica Stiftung grant 84803, no website. The funders had no role in study design, data collection and analysis, decision to publish, or preparation of the manuscript. There was no additional external funding received for this study.

**Competing interests:** The authors have declared that no competing interests exist.

framework for understanding *agr*-type-specific regulatory strategies and their potential roles in *S. aureus* pathogenesis.

## Introduction

*Staphylococcus aureus* is a commensal bacterium colonizing the human skin and nares [1], as well as a human pathogen causing life-threatening infections [2,3]. Its pathogenic potential is largely attributed to virulence factors, tightly regulated by networks, such as the accessory gene regulator (*agr*) system [4]. The *agr* is a quorum-sensing, two-component regulatory system classified into four types, based on *agr*-locus (*agrBCDA*) polymorphisms, affecting predominantly the autoinducing peptide (AIP) encoded by *agrD* and its sensor AgrC. The *agr*-locus is structured around two key promoters: P2, which governs expression of the *agrBDCA* operon, and P3, which controls production of RNAIII, the main effector of the system [5]. Each strain harbors a single *agr*-locus, and each clonal complex (CC) or sequence type (ST) typically encompass only one *agr*-type [6]. Previous studies reported associations between *agr*-type and infection severity [7–9]. However, assessing the specific contribution of *agr*-type to virulence is challenging due to the confounding effects of genetic background. To date, only two studies have compared *agr*-types within an identical genetic background, focusing on a limited set of *agr*-regulated targets [10,11]. To gain a comprehensive understanding of the impact of the *agr*-type on the global transcriptome, we performed RNA sequencing at multiple growth phases on congenic *S. aureus* strains representing each *agr*-type (*agr*-I, *agr*-II, *agr*-III, *agr*-IV) and a knock-out mutant (Δ*agr*) in the same genetic background (Newman) [11]. Despite equal bacterial density, major differences in the kinetics of *agr*-activation between *agr*-I/IV and *agr*-II/III were observed, leading to differential expression of virulence factors and metabolic pathways.

## Materials and methods

### Bacterial strains and growth conditions

*S. aureus* congenic strains Δ*agr*, *agr*-I, *agr*-II, *agr*-III and *agr*-IV, were previously constructed in the Newman genetic background by deletion of the native *agr*-locus and then complemented with any of the four *agr*-types [11]. Strains were cultured overnight in tryptic soy broth at 37°C, with shaking. Then, strains were subcultured 1:20 for 1h30 three times with double centrifugation-washing with phosphate-buffered saline in-between. The last subculture was incubated for 12h, and samples were collected at 1h30, 6h, and 12h. Bacteria were harvested by centrifugation and stored in RNAlater (Sigma Aldrich) prior to RNA extraction.

### RNA extraction and sequencings

Bacterial pellets were resuspended in 10 mM Tris (pH8) with lysostaphin (Biosynth, 100 μg/mL), incubated for 1h at 4°C and 10 min at 37°C. Total RNA was extracted using the RNeasy Plus Mini Kit (Qiagen), followed by double TURBO DNase treatment

(Invitrogen). RNA quality and concentration were assessed using the RNA ScreenTape Assay and TapeStation System (Agilent). Library preparation (TruSeq RNA Library Prep, Illumina) and paired-end 150 bp sequencing were performed by the Functional Genomics Center Zurich on an Illumina NovaSeq X Plus, yielding $\sim$ 24 million reads per sample.

### RNA-seq and statistical analysis

Reads were processed using a custom pipeline: https://zenodo.org/records/18492655. The steps included quality control and filtering, rRNA removal (SortMeRNA 4.3.7) and alignment to the *S. aureus* Newman reference genome (RefSeq GCF_000010465.1, bowtie2 2.5.2) and read summarization. DE analysis, ORA and GSEA were conducted in R (v4.5.2) using DESeq2 (v1.50.2, [12]) and clusterProfiler (v4.18.3, [13]). Full analysis scripts and RNAseq count files are available at: https://zenodo.org/records/19554265. Comparisons of *agrA* and RNAIII relative expression to *gyrB* were performed using GraphPad Prism (v10.6.1).

### Artificial intelligence tools and technologies

The manuscript was written by the authors. Microsoft M365 Copilot (GPT-5.2, last accessed on 14th of April 2026) was employed for grammar correction, text streamlining and improving language clarity, and as a tool to explore alternative phrasings. Github Copilot (Claude Sonnet 4.5, last accessed at 14th of April 2026) was used for code readability and efficiency improvements and streamlining. The authors have thoroughly reviewed, verified, and edited any passages generated by any LLM, taking full responsibility for the manuscript and overall quality and accuracy.

## Results

### *Agr*-activation kinetics differ among *agr*-types

To assess specifically, how the *agr*-types influence *S. aureus* transcriptome, we used congenic strains, sequentially subcultured, to reset *agr*-activation and to reduce AIP accumulation. Samples were collected at 1h30 (early exponential phase), 6h (late exponential phase) and 12h (stationary phase). Since *agr*-activation in bacterial population is density-dependent, we confirmed that all strains grew similarly and had a comparable density (Fig 1A).

We then assessed *agrA* and RNAIII transcription reflecting the *agr*-activation state, across strains and time points (Fig 1B-C and Panels A-B S1 Fig). Across the three time points, the four *agr*-types showed distinct expression kinetics. At 1h30, all strains had low expression levels for both genes (Fig 1B-C). By 6h, *agr*-I and *agr*-IV showed high and significant expression of *agrA* and RNAIII, *agr*-II displayed an intermediate induction (significant RNAIII increase compared to 1h30) while *agr*-III showed no detectable induction (Panels A-B S1 Fig). At 12h, *agr*-I, *agr*-II, and *agr*-IV displayed high *agrA* and RNAIII expression, while *agr*-III reached intermediate expression levels, comparable with *agr*-II at 6h (Fig 1B–C). Both gene expressions were significantly higher in *agr*-I, *agr*-II and *agr*-IV compared to *agr*-III (Fig 1B-C), and overall, only *agrA* expression in *agr*-III increased significantly between 1h30 and 12h (Panel A S1 Fig), consistent with a delayed activation. Overall, despite similar growth kinetics, *agr*-I and *agr*-IV activated earlier than *agr*-II, while *agr*-III did not show strong *agr*-activation compared to the other *agr*-types, even after 12h of growth.

### *Agr*-activation shapes the transcriptomes

To evaluate *agr*-type-dependent transcriptome variation, we performed a principal component analysis (PCA) on the full dataset (three biological replicates for each time point), omitting the *agr*-locus to avoid $\Delta agr$-driven bias (Fig 2). Samples clustered primarily by time point, with strong separation between early exponential 1h30 and both 6h and 12h time points (Fig 2A and Panel A S2 Fig).

Next, we focused on each individual time point for further analysis. We did not find any clustering at 1h30 (Fig 2B and Panel B S2 Fig). For both 6h and 12h, the PC1 axis explained most of the variance, with 56% and 51% respectively (Fig

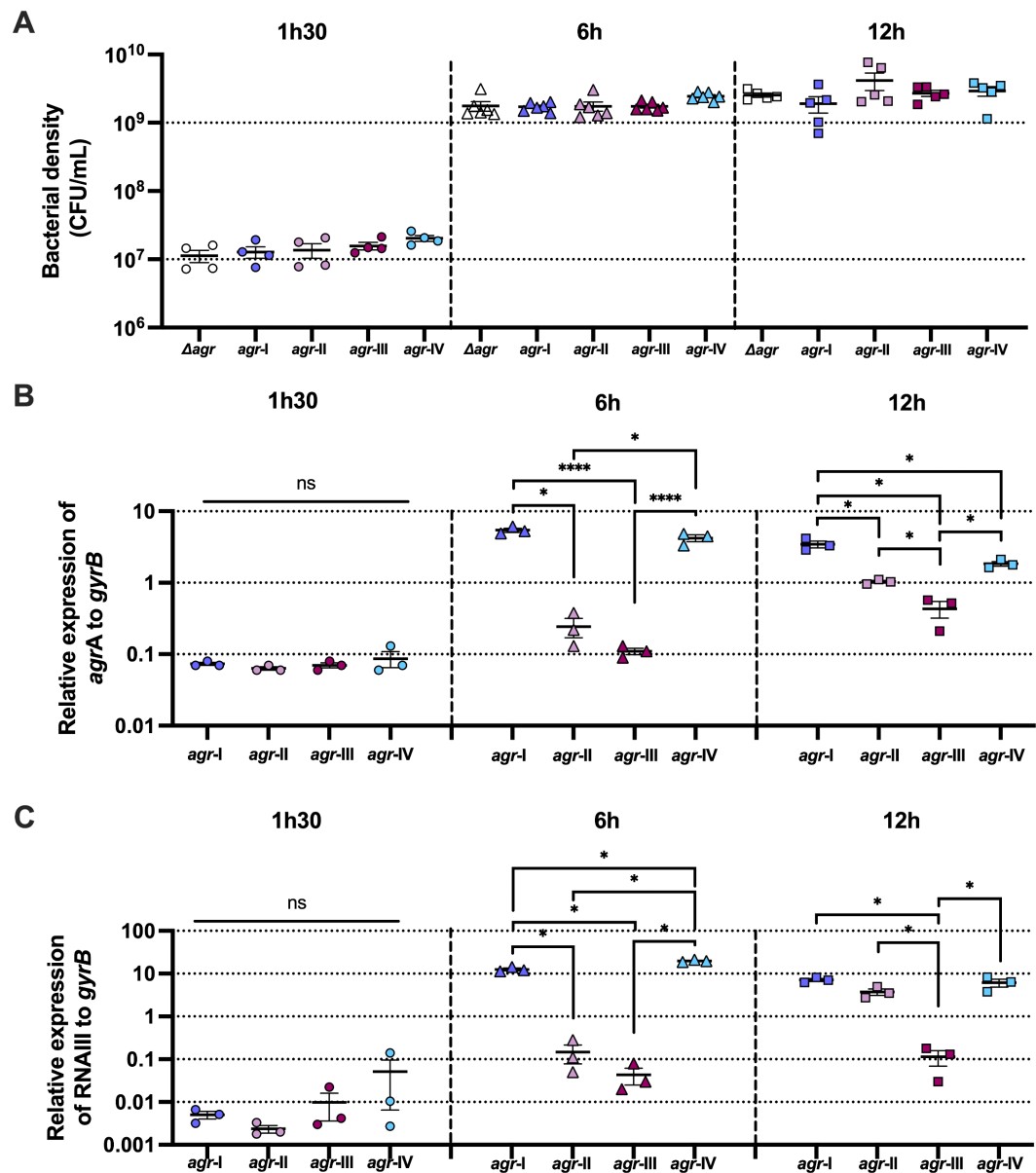

**Fig 1. Differential *agr*-activation kinetics of the four *agr*-types.** A. Bacterial density measured as CFU/mL for each strain and time point. Error bars represent the standard error of the mean (SEM). A minimum of four biological replicates were performed. B and C. Relative expression of *agrA* (B) and RNAIII (C) to *gyrB* using RNAseq counts (number of reads of *agrA* or RNAIII / number of reads of *gyrB*) for each strain except Newman$\Delta$*agr*, as no reads could be detected for neither *agrA* or RNAIII. Error bars represent the standard error of the mean (SEM). Two-way ANOVA with interaction term of *agr*-type and timepoint on log10 relative expression and Turkey's multiple comparisons tests were performed; adjusted p-value < 0.0001 - ****, < 0.05 - *; ns – not significant. Three biological replicates were performed.

2C-D and Panels C-D S2 Fig), whereas the PC2 axis only explained 18% and 10% (Fig 2C-B). Hence, three separate clusters of *agr*-I, *agr*-IV and all other strains were observed at 6h (Fig 2C and Panel C S2 Fig). At 12h, three different clusters were found: i) *agr*-I with *agr*-IV, ii) *agr*-II samples displaying an intermediate profile, and iii) *agr*-III clustering with $\Delta$*agr* (Fig 2D and Panel D S2 Fig). Although variability within the clusters along the PC2 was observed, these findings, in regard

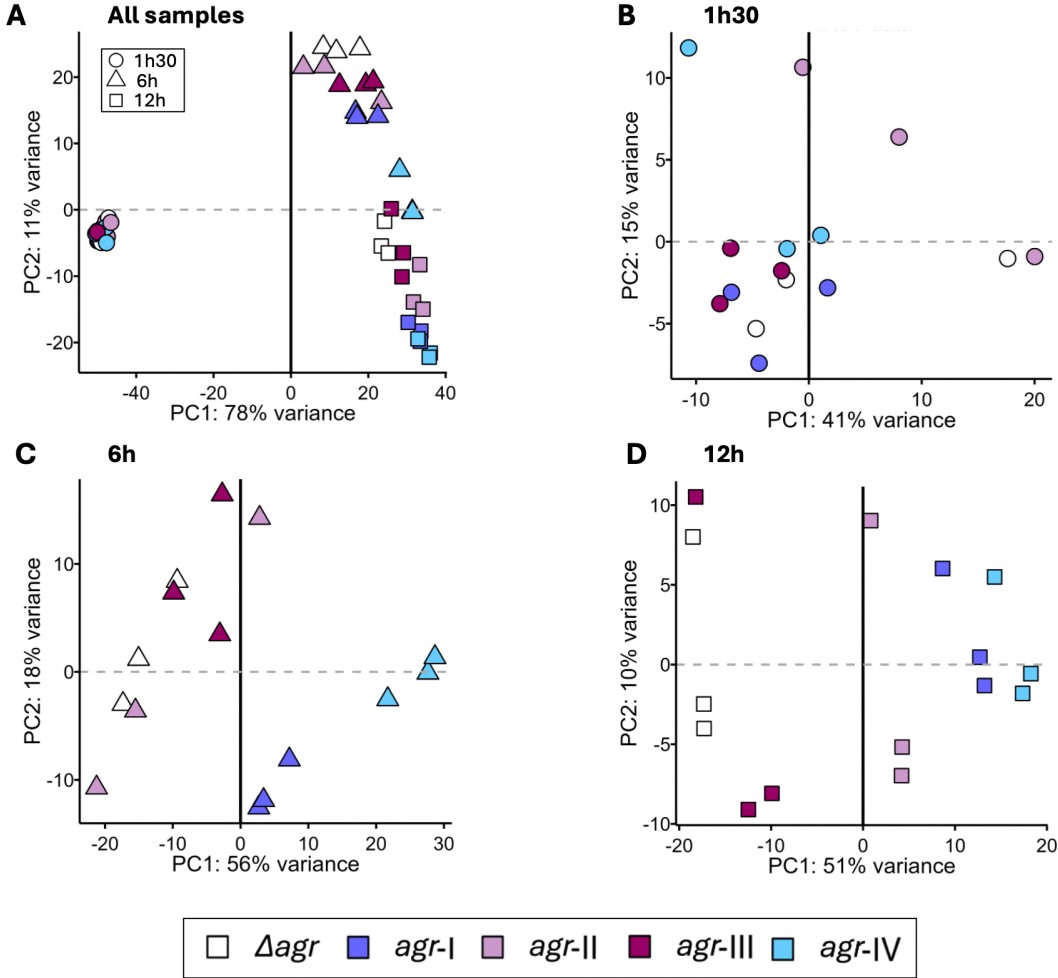

**Fig 2. Principal component analysis clustering driven by the *agr*-activation profile.** Dot-plots of the principal component analysis (PCA), with PC1 and PC2 axes, using the entire transcriptome beside the *agr*-locus, with all time points in A, at time point 1h30 in B, 6h in C and 12h in D. Solid (PC1) and dashed (PC2) lines mark the zero score for each principal component.

with identified *agr*-activation profiles (Fig 1), suggest that the transcriptome is driven by both time and the *agr*-activation profile, rather than the *agr*-type alone, despite identical genetic background and comparable bacterial density of the cultures.

### *Agr*-activation triggers specific virulence and metabolic profiles

To investigate how *agr*-type shapes the different transcriptomes during growth, we first compared each *agr*-type to the Δ*agr* strain. At 1h30, none of the strains showed differential expression (DE) relative to Δ*agr* (S3 Fig and S1 Table), consistent with the PCA clustering (Fig 2B) and the low *agr*-activity at this time point (Fig 1B-C). At 6h, *agr*-I and *agr*-IV displayed strong upregulation of all phenol-soluble modulin (*psm*) genes (Fig 3A and 3D). *Agr*-IV showed the most extensive transcriptomic shift, with 233 up- and 139 downregulated genes, relative to Δ*agr* (S2 Table). Upregulated loci included the capsule operon and several key virulence factors such as *hlgCB*, *lukDE*, multiple serine proteases, whereas many superantigen-like genes (*ssl*) and amino-acid metabolism pathways were downregulated (Fig 3D and S2

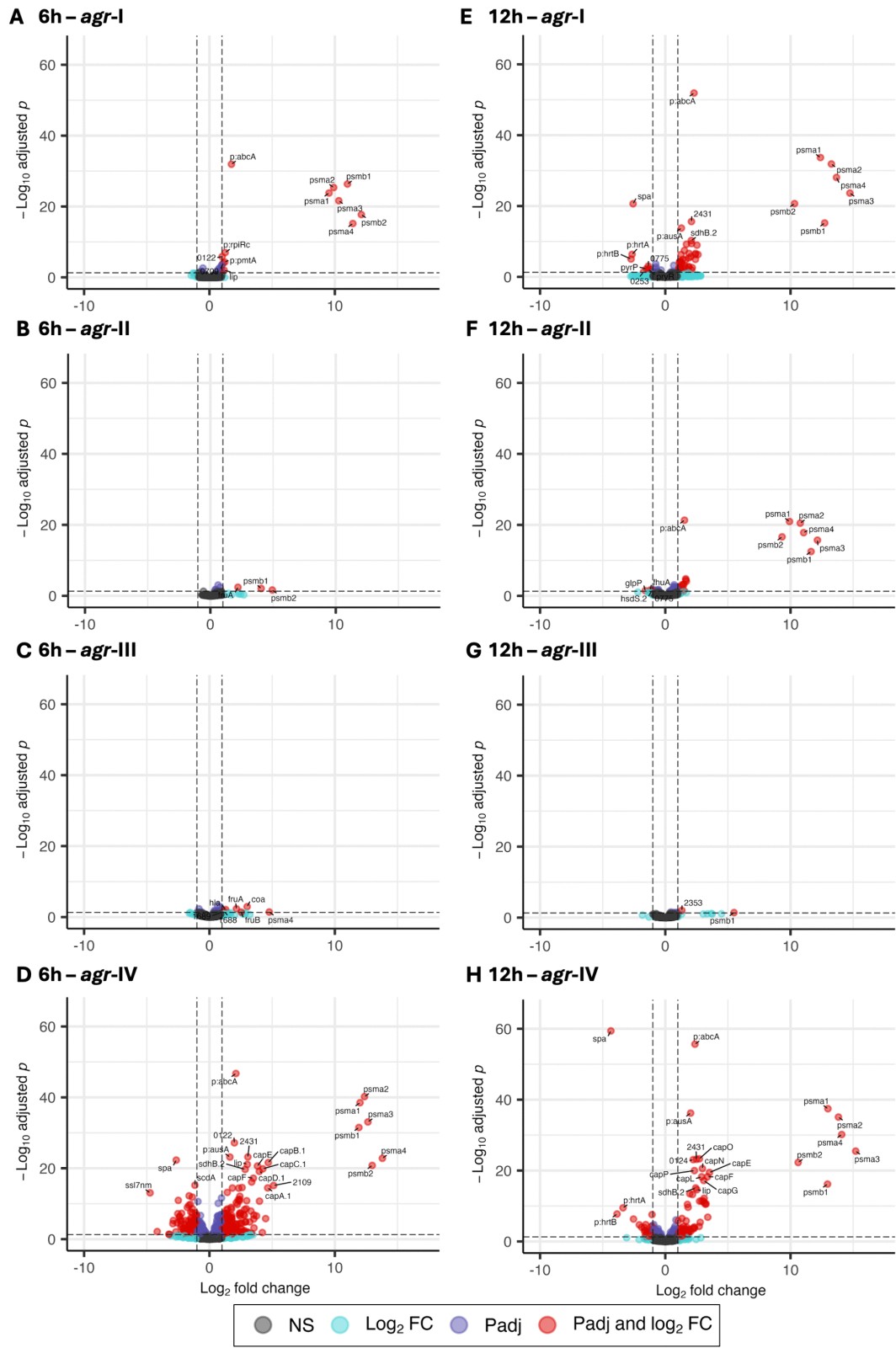

**Fig 3. Global transcriptomes differ depending on the *agr*-type and *agr*-activation.** Volcano plots of the pairwise comparisons of the four *agr*-types compared to Δ*agr*, at 6h (A-D) and 12h (E-H). Comparisons to *agr*-I in A and E, to *agr*-II in B and F, to *agr*-III in C and G, and to *agr*-IV in D and H.

Significantly differentially expressed genes with a fold change of |2| (log2 fold change (LFC) of 1) and adjusted p-value < 0.05 are in red. Dashed lines indicate fold change threshold and adjusted p-value threshold (0.05). Gene names of top 15 genes are annotated according to Aureowiki Newman gene annotation; if only pangenome annotation was available "p:" prefix was added to the gene's name, for genes without gene's name the four digits following the "NWMN" tag were kept as gene ID. ns: not significant, FC: fold change, Padj: adjusted p-value.

Table). In contrast, *agr*-II and *agr*-III showed only minimal changes, primarily low induction of *psm* genes, reflecting their limited *agr*-activation at this stage (Fig 3B-C and S2 Table). At 12h, *agr*-III remained transcriptionally similar to Δ*agr* (Fig 3G), consistent with its slow *agr*-activation. In comparison, *agr*-I, *agr*-II, and *agr*-IV strongly overexpressed all *psm* genes (Fig 3E, F, H and S3 Table). Capsule expression remained high in *agr*-IV and expression increased in *agr*-I as well (Fig 3E, H and S3 Table). Furthermore, *agr*-IV displayed the broadest profile of DE, including unique downregulation of purine and pyrimidine metabolism genes (Fig 3H), while *hlgCB*, *lukDE*, and protease genes were not significantly induced at 12h anymore.

After characterizing variation relative to Δ*agr*, we analyzed the pairwise relationships within the *agr*-type strains *agr*-I, -II, -III and -IV. At 1h30, differences were limited, with only 30 significant DE genes with a fold change higher of |2| (S1 Table). At 6h, 411 significant DE genes (fold change of |2|) were found (S2 Table) and most DE genes (fold change of |4|) fell into three functional categories: virulence factors, capsule genes, and metabolic pathways (riboflavin, fructose, gluconate, S4 Fig). *Agr*-IV upregulated capsule and *psm* genes stronger than other *agr*-types, while *ssl* genes were downregulated in *agr*-IV but upregulated in *agr*-II and *agr*-III. At 12h, transcriptional differences between *agr*-types were weaker overall (149 DE genes), though *agr*-IV maintained high capsule and *psm* expression, matched by *agr*-I and, to a lower level, by *agr*-II (S3 Table). *Agr*-III displayed the lowest variability across time (S4 Fig and S3 Table). Over-representation analysis (ORA) recapitulated these patterns (Table 1). Capsule biosynthesis was the most enriched pathway in *agr*-IV at both 6h and 12h. Virulence-related pathways, including toxins and secreted proteases, were likewise enriched among upregulated genes in *agr*-IV. Notably, when comparing *agr*-III and *agr*-IV at 6h, the "*S. aureus* infection" pathway (sae05150) appeared as both up- and down-regulated gene sets with distinct genes in each (Table 1). This indicates that *agr*-III is not simply avirulent but expresses an alternative virulence gene panel, shaped by a delayed *agr*-activation. ORA also identified enrichment of metabolic pathways. At 6h, phosphotransferase system (PTS), fructose and histamine metabolisms were enriched in all *agr*-types compared to *agr*-I. While at 12h, nucleotides and arginine metabolisms were differently enriched depending on the *agr*-type, highlighting that *agr*-type influences not only virulence but broader cellular physiology.

To detect coordinated transcriptional changes missed by single-gene thresholds, we performed Gene Set Enrichment Analysis (GSEA) (S4 Table). GSEA showed early enrichment of metabolic pathways in *agr*-II and *agr*-IV at 1h30 and 6h compared to *agr*-I (S4 Table, sheets 1 and 2), while *agr*-III showed broader metabolic enrichment only at 12h (S4 Table, sheet 3), indicating its delayed *agr*-activation, linked to differential metabolism compared to the other *agr*-types. *Agr*-I and *agr*-IV, which exhibited the strongest *agr*-activation, showed fewer stationary-phase metabolic changes at 12h, consistent with a shift toward virulence-focused genes expression (S4 Table, sheet 3). Together, DE analysis, ORA and GSEA reveal that *agr*-type not only determines the timing and magnitude of core *agr*-regulated virulence genes but also drives distinct, temporally resolved metabolic pathways during growth.

## Discussion

The *agr*-system is one of the most studied quorum-sensing systems and the most important virulence regulator in *S. aureus* [4], yet its strong linkage to genetic background has hindered efforts to investigate *agr*-type-specific effects. Using congenic strains sharing the same genetic background and exhibiting similar growth dynamics, we demonstrate that different *agr*-types possess distinct *agr*-activation kinetics, resulting in clearly separable temporal transcriptomic profiles.

**Table 1. Significant pathways enriched among up- and downregulated genes from over-representation analysis (ORA).**

| Comparison | Enriched pathway | Gene set direction | Gene Ratio[a] | Genes | BH adjusted p-value |
|---|---|---|---|---|---|
| **1h30** | | | | | |
| agr-II vs III | *Staphylococcus aureus* infection | Up-regulated | 3/7 | sdrD, lukG, lukH | 0.01436 |
| agr-II vs IV | *Staphylococcus aureus* infection | Up-regulated | 3/4 | sdrC, sdrD, hlgC | 0.00044 |
| **6h** | | | | | |
| agr-I vs II | Fructose and mannose metabolism | Down-regulated | 2/4 | fruA, fruB | 0.00552 |
| | Phosphotransferase system (PTS) | Down-regulated | 2/4 | fruA, fruB | 0.00628 |
| | Quorum-sensing | Up-regulated | 5/7 | psmα1–3, psmβ1–2 | 7.1549e-05 |
| agr-I vs III | Phosphotransferase system (PTS) | Down-regulated | 4/5 | NWMN_0323, fruB, fruA, p:glvC | 8.4247e-06 |
| | Fructose and mannose metabolism | Down-regulated | 2/5 | fruA, fruB | 0.00798 |
| | Microbial metabolism in diverse environments | Down-regulated | 3/5 | pflB, NWMN_0323, fruA | 0.04895 |
| | Quorum-sensing | Up-regulated | 5/7 | psmα1–3, psmβ1–2 | 0.00019 |
| agr-I vs IV | Biosynthesis of various nucleotide sugars | Down-regulated | 5/78 | capD.1, capE, capF, capG, capO | 0.00556 |
| | Phosphotransferase system (PTS) | Down-regulated | 8/78 | NWMN_0323, NWMN_0324, p:murP, ulaA, fruB, fruA, lacE, p:glvC | 0.00389 |
| | Histidine metabolism | Down-regulated | 8/78 | NWMN_2026, aldA, hutI, hutU, hutG, hisIE, hisF, hisA | 0.00018 |
| | Quorum-sensing | Down-regulated | 16/78 | psmα1–3, comK, p:nikD, ribD, splF, splE, splD, splA, bsaG, bsaE, bsaF, bsaP, bsaA1, p:secA2 | 0.00026 |
| | *Staphylococcus aureus* infection | Up-regulated | 7/25 | spa, ssl7nm, isdA, p:flr, p:efb, p:scc, chp | 0.00050 |
| agr-II vs IV | Biosynthesis of various nucleotide sugars | Down-regulated | 5/78 | capD.1, capE, capF, capG, capO | 0.00979 |
| | Quorum-sensing | Down-regulated | 21/78 | psmα1–3, psmβ1–2, comK, sspA, nikB, ribD, splF, splE, splD, splA, bsaG, bsaE, bsaF, basP, bsaA2, bsaA1, kdpE, secA2 | 5.6976e-08 |
| | *Staphylococcus aureus* infection | Up-regulated | 10/56 | spa, ssl6nm, ssl7nm, ssl11nm, isdA, p:flr, p:efb, p:scc, chp, clfB | 0.00092 |
| agr-III vs IV | Biosynthesis of various nucleotide sugars | Down-regulated | 6/47 | capD.1, capE, capF, capG, capO, capP | 1.8628e-05 |
| | Biosynthesis of nucleotide sugars | Down-regulated | 6/47 | capD.1, capE, capF, capG, capO, capP | 0.00205 |
| | *Staphylococcus aureus* infection | Down-regulated | 8/47 | NWMN_0851, clfA, lukD, lukE, lukG, hlgA, hlgC, hlgB | 0.00205 |
| | Quorum-sensing | Down-regulated | 14/47 | psmα1–3, psmβ1–2, comK, sspA, splF, splE, splD, splA, bsaP, bsaA2, bsaA1 | 9.2178e-06 |
| | Phosphotransferase system (PTS) | Up-regulated | 4/17 | glcA, treP, mtlF, ptsG | 0.00378 |
| | *Staphylococcus aureus* infection | Up-regulated | 6/17 | spa, ssl7nm, ssl11nm, p:efb, p:scc, chp | 0.00037 |
| **12h** | | | | | |
| agr-I vs II | Biosynthesis of various nucleotide sugars | Up-regulated | 5/13 | capD.1, capE, capF, capG, capP | 5.7310e-07 |
| | Biosynthesis of nucleotide sugars | Up-regulated | 5/13 | capD.1, capE, capF, capG, capP | 2.1498e-05 |

*(Continued)*

**Table 1.** (Continued)

| Comparison | Enriched pathway | Gene set direction | Gene Ratio[a] | Genes | BH adjusted p-value |
|---|---|---|---|---|---|
| agr-I vs III | Biosynthesis of various nucleotide sugars | Up-regulated | 6/19 | capD.1, capE, capF, capG, capO, capP | 8.1555e-08 |
| | Biosynthesis of nucleotide sugars | Up-regulated | 6/19 | capD.1, capE, capF, capG, capO, capP | 9.3153e-06 |
| | Quorum-sensing | Up-regulated | 5/19 | $psm\alpha1$–3, $psm\beta1$–2 | 0.03955 |
| agr-I vs IV | Biosynthesis of various nucleotide sugars | Down-regulated | 4/5 | capF, capG, capO, capP | 1.2026e-07 |
| | Biosynthesis of nucleotide sugars | Down-regulated | 4/5 | capF, capG, capO, capP | 1.6987e-06 |
| | Quorum-sensing | Up-regulated | 5/19 | $psm\alpha1$–3, $psm\beta1$–2 | 0.03955 |
| | *Staphylococcus aureus* infection | Up-regulated | 1/19 | spa | 0.03952 |
| agr-II vs III | Quorum-sensing | Up-regulated | 5/9 | $psm\alpha1$–3, $psm\beta1$–2 | 0.00058 |
| agr-II vs IV | Biosynthesis of various nucleotide sugars | Down-regulated | 6/19 | capD.1, capE, capF, capG, capO, capP | 6.6727e-08 |
| | Biosynthesis of nucleotide sugars | Down-regulated | 6/19 | capD.1, capE, capF, capG, capO, capP | 7.6216e-06 |
| | Arginine biosynthesis | Down-regulated | 3/19 | arcC.2, arcB, arcA | 0.03140 |
| | Nucleotide metabolism | Up-regulated | 4/15 | p:xpt, guaB, guaA, pryR | 0.02090 |
| agr-III vs IV | Biosynthesis of various nucleotide sugars | Down-regulated | 6/33 | capD.1, capE, capF, capG, capO, capP | 3.6039e-06 |
| | Biosynthesis of nucleotide sugars | Down-regulated | 6/33 | capD.1, capE, capF, capG, capO, capP | 0.00036 |
| | Arginine biosynthesis | Down-regulated | 5/33 | ureA, ureC, arcC.2, arcB, arcA | 0.00246 |
| | Pyrimidine metabolism | Up-regulated | 5/27 | p:psuG, pryR, pyrC, pyrAA, pyrAB | 0.00563 |
| | Purine metabolism | Up-regulated | 6/27 | p:xpt, guaB, guaA, purE, purK, purC | 0.00563 |

Genes were first filtered by differential expression analysis (DESeq2) using an adjusted p-value threshold of 0.05 and fold change of |2| (LFC of 1). These filtered genes were then used for ORA using the KEGG sae (T00557) database and parameters minGSSize of 10 and maxGSSize of 500. Pathways were considered significantly enriched based on both adjusted p ≤ 0.05 (Benjamini-Hochberg, BH) and FDR corrected q-value cutoff of 0.2. Gene set direction refers to the first strain mentioned in the comparison. GeneRatio = overlap size (k) / size of query gene list (n).

Early *agr*-activation in *agr*-I and *agr*-IV drove strong induction of PSMs, capsule biosynthesis, and several pore-forming toxins, whereas *agr*-II activation was more gradual and *agr*-III activated only in stationary growth-phase, resulting in markedly different virulence signatures in early, late and stationary growth-phase. Importantly, *agr*-II and *agr*-III did not exhibit an avirulent profile; instead, they expressed alternative virulence factors focusing on superantigen-like genes during early stationary phase. Furthermore, adhesion factors such as *spa* or *fnbA*, that are expected to be downregulated in *agr*-activated bacteria [14] showed a similar profile of downregulation in our experiments for *agr*-I and *agr*-IV. This alternative virulence profile of *agr*-III aligns with clinical observations that both *agr*-III or *agr*-deficient strains frequently cause invasive infections [15,16], indicating reliance on virulence repertoires less dependent on directly *agr*-regulated toxins. Unexpectedly, the adhesion factor *clfA* was upregulated at 6h in *agr*-IV, despite sharing regulatory features with *spa* and *fnbA*, highlighting the complexity and partial uncoupling within the *agr*-regulatory network. The transient reduction of *chp* expression in *agr*-IV despite rapid *agr*-activation is consistent with previous reports showing limited *agr*-control over *chp* and reflects its integration into a broader regulatory network involving SaeRS, SarA-family regulators, and stress-responsive pathways [17,18].

*Agr*-IV displayed the strongest and fastest *agr*-activation, as previously observed in another genetic background [19]. This correlated with the broadest transcriptional response, including early, high-level capsule expression and differential modulation of metabolic pathways. These findings suggest that the *agr*-type not only affects virulence factors but also impacts the metabolic strategies used to adapt to the same environment. The arginine deiminase (ADI) system as well as the *kdp* and *ure* operons were upregulated in *agr*-IV at 12h. All three are involved acid stress response [20]. In addition, the ADI system specifically has been shown to be modulated by available carbon sources [21,22], and our data show that *agr*-IV upregulated capsule production early, which is known to be strongly glucose dependent [23,24]. Further research is needed to confirm this hypothesis. Our experiments have been performed only in the Newman genetic background in triplicates. Similar studies with different genetic backgrounds to unravel co-evolutionary adaptation of genetic background and the *agr*-type as well as increased replicates of transcriptomics analysis can further strengthen the robustness of our observations. However, our study suggests that such *agr*-type-specific transcriptional architectures may affect the ecological success of specific *agr*-lineages.

Previous studies found correlations between *agr*-type or the CC/ST with specific infections [7,8], notably *agr*-III and the CC30 (*agr*-III) with the *tst* carriage and the toxic shock syndrome [19,25]. However, some recent studies show that no specific associations were found between the *agr*-type and pathogenicity [16] or specific toxin carriage [26], underlying the complexity of *S. aureus* genetic impact on its virulence potential.

We demonstrated that the *agr*-type influenced the transcriptomic profile in congenic strains and under controlled growth conditions. However, the underlying molecular mechanisms, whether rooted in promoter architecture, AgrC signaling, or peptide-receptor coevolution, remain to be elucidated.

## Supporting information

**S1 Fig. Differential *agr*-activation kinetics of the four *agr*-types.** A and B. Relative expression of *agrA* (A) and RNAIII (B) to *gyrB* using RNAseq counts (number of reads of *agrA* or RNAIII / number of reads of *gyrB*) for each strain except Newman $\Delta agr$, as no reads could be detected for neither *agrA* nor RNAIII (same data as in Fig.1). Error bars represent the standard error of the mean (SEM). Two-way ANOVA tests on log10 transformed relative expressions and Turkey's multiple comparisons tests were performed; adjusted p-value < 0.0001 - ****, < 0.001 - ***, < 0.01 - **, < 0.05 - *; ns – not significant. Three biological replicates were performed.
(TIF)

**S2 Fig. Cluster identification from PCA analysis.** Dot-plots of the principal component analysis (PCA), using PC1 axis, depending on the *agr*-type, using the entire transcriptome beside the *agr*-locus, with all time points in A, at time point 1h30 in B, 6h in C and 12h in D. Standard error of the mean (SEM) for each time point in A, and per strain in B, C and D, are represented in black.
(TIF)

**S3 Fig. No expression differences in the absence of *agr*-activation.** Volcano plots of pairwise comparisons of the four *agr*-types to $\Delta agr$ at 1h30. Comparisons to *agr*-I in A, to *agr*-II in B, to *agr*-III in C and to *agr*-IV in D. Dashed lines indicate fold change threshold (fold change of |2| (LFC of 1)) and adjusted p-value threshold (0.05). NS: not significant, FC: fold change.
(TIF)

**S4 Fig. Genes differentially expressed in pairwise *agr*-type comparisons.** Heatmaps of regularized log counts (normalized raw counts and input for DESeq2 analysis) of the genes differentially expressed for at least one pairwise comparison between the four *agr*-types (S2 and S3 Tables), with a fold change of |4| (LFC of 2)). 94 genes out of the 411 at 6h (A), and 43 out of the 149 genes at 12h (B) were included (no genes with a fold change of |4| (LFC of 2)) were

differentially expressed at 1h30 out of the 30 genes). Biological replicates for each strain are shown. Biological functions from KEGG database and detected in over-representation analysis (ORA, •) or described in the literature to belong to these pathways (○) are indicated next to the gene names. Gene names are annotated according to Aureowiki Newman gene annotation; if only pangenome annotation was available "p:" prefix was added to the gene's name, for genes without gene's name the four digits following the "NWMN" locus-tag were kept as gene ID.
(TIF)

**S1 Table. All genes significantly differentially expressed for at least one pairwise comparison at 1h30.** Blue: genes with a fold change (FC) <−2; red: genes with a FC > 2; green: adjusted p-value (p-adj) <0.05. Gene names are annotated according to Aureowiki Newman gene annotation; if only pangenome annotation was available "p:" prefix was added to the gene's name, for genes without gene's name the four digits following the "NWMN_" tag were kept as gene ID. Negative Binomial GLM with Wald test for pairwise comparisons were performed with Benjamini–Hochberg correction.
(XLSX)

**S2 Table. All genes significantly differentially expressed for at least one pairwise comparison at 6h.** Blue: genes with a fold change (FC) <−2; red: genes with a FC > 2; green: adjusted p-value (p-adj) <0.05. Gene names are annotated according to Aureowiki Newman gene annotation; if only pangenome annotation was available "p:" prefix was added to the gene's name, if only the symbol annotation was available "s:" prefix was added to the gene's name, for genes without gene's name the four digits following the "NWMN_" tag were kept as gene ID. Negative Binomial GLM with Wald test for pairwise comparisons were performed with Benjamini–Hochberg correction.
(XLSX)

**S3 Table. All genes significantly differentially expressed for at least one pairwise comparison at 12h.** Blue: genes with a fold change (FC) <−2; red: genes with a FC > 2; green: adjusted p-value (p-adj) <0.05. Gene names are annotated according to Aureowiki Newman gene annotation; if only pangenome annotation was available "p:" prefix was added to the gene's name, if only the symbol annotation was available "s:" prefix was added to the gene's name, for genes without gene's name the four digits following the "NWMN_" tag were kept as gene ID. Negative Binomial GLM with Wald test for pairwise comparisons were performed with Benjamini–Hochberg correction.
(XLSX)

**S4 Table GSEA results for all *agr*-type pairwises comparisons.** Ranking metric prioritizes the Wald statistic (res$stat) of DEseq2, pvalueCutoff of 1 was used (no gene selection from DE), Benjamini–Hochberg FDR correction applied, and pathways with adjusted p-value <0.05 retained. minGSSize of 10 and maxGSSize of 500 were used as default parameters. Positively- or negatively-enriched refers to the first strain mentioned in the comparison. Enriched terms are reported with normalized enrichment scores (NES), p-values, and FDR q-values.
(XLSX)

## Acknowledgments

We thank Bo Shopsin for providing the *agr*-congenic Newman strains. RNA sequencing was performed at the Functional Genomics Center Zurich (FGCZ) of University of Zurich and ETH Zurich. We thank Alejandro Gómez Mejia, Federica Andreoni and Andrea Tarnutzer for their help on scientific discussions and manuscript revision.

## Author contributions

**Conceptualization:** Mariane Pivard, Julian Bär.

**Data curation:** Tomas Demeter.

**Formal analysis:** Mariane Pivard, Julian Bär, Tomas Demeter.

**Funding acquisition:** Annelies S. Zinkernagel.

**Investigation:** Mariane Pivard, Julian Bär, Srikanth Mairpady Shambat.

**Methodology:** Mariane Pivard, Julian Bär, Srikanth Mairpady Shambat.

**Project administration:** Annelies S. Zinkernagel.

**Software:** Tomas Demeter.

**Validation:** Mariane Pivard, Julian Bär, Srikanth Mairpady Shambat, Annelies S. Zinkernagel.

**Visualization:** Mariane Pivard, Julian Bär.

**Writing – original draft:** Mariane Pivard, Julian Bär, Srikanth Mairpady Shambat, Annelies S. Zinkernagel.

**Writing – review & editing:** Mariane Pivard, Julian Bär, Srikanth Mairpady Shambat, Annelies S. Zinkernagel.

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
