## [Decision Letter · Decision Letter 0]

6 Apr 2026

PONE-D-26-11159*Staphylococcus aureus  agr*-type vs genetic background: molecular signatures determining differential metabolism and virulence potentialPLOS One

Dear Dr. Pivard,

Thank you for submitting your manuscript to PLOS ONE. After careful consideration, we feel that it has merit but does not fully meet PLOS ONE’s publication criteria as it currently stands. Therefore, we invite you to submit a revised version of the manuscript that addresses the points raised during the review process. Please read and answer carefully all the points  raised by the two reviewers.

We look forward to receiving your revised manuscript.

Kind regards,

Herminia de Lencastre, Ph.D.

Academic Editor

PLOS One

Journal Requirements:

[A.S.Z. was supported by the Swiss National Science Foundation (SNSF; https://www.snf.ch/en) project grant 310030_204343 and M.P. by the Promedica Stiftung grant 84803.].

4. Please remove any funding-related text from the manuscript. Funding information should not appear in any section or other areas of your manuscript. We will only publish funding information present in the Funding Statement section of the online submission form.

5. We noted in your submission details that a portion of your manuscript may have been presented or published elsewhere.

[A subset of the RNA-seq data presented in this manuscript—specifically, the 1h30 time point for agr types I, II, III, and IV (excluding the Δagr strain)—is also included in a separate manuscript that investigates agr-activation at the single‑cell level using microfluidic microscopy. In that other work, the RNA-seq dataset appears only in Supplementary Figure S5, where it is used to assess whether differences in AIP sensitivity among agr-types could be attributed to variation in agr-regulators or agr-regulated genes.

The purpose, analyses, and conclusions drawn from these RNA-seq data in the other manuscript are entirely distinct from those reported here. The use of the dataset in the present study addresses different biological questions and does not overlap in scope or interpretation with the other publication.]

Please clarify whether this publication was peer-reviewed and formally published. If this work was previously peer-reviewed and published, in the cover letter please provide the reason that this work does not constitute dual publication and should be included in the current manuscript.

6. When completing the data availability statement of the submission form, we note that you have provided the source of datasets that is used for your study. However, the accession number that you provided shows “record not found”, which suggests that it is currently set on private, and will be possibly available on acceptance.

We strongly recommend all authors decide on a data sharing plan before acceptance, as the process can be lengthy and hold up publication timelines. Please note that, though access restrictions are acceptable now, your entire data will need to be made freely accessible if your manuscript is accepted for publication. This policy applies to all data except where public deposition would breach compliance with the protocol approved by your research ethics board. If you are unable to adhere to our open data policy, please kindly revise your statement to explain your reasoning and we will seek the editor's input on an exemption. Please be assured that, once you have provided your new statement, the assessment of your exemption will not hold up the peer review process.

Reviewers' comments:

Reviewer's Responses to Questions

**Comments to the Author**

1. Is the manuscript technically sound, and do the data support the conclusions?

Reviewer #1: Yes

Reviewer #2: Yes

2. Has the statistical analysis been performed appropriately and rigorously? 

Reviewer #1: I Don't Know

Reviewer #2: Yes

3. Have the authors made all data underlying the findings in their manuscript fully available?

Reviewer #1: Yes

Reviewer #2: Yes

4. Is the manuscript presented in an intelligible fashion and written in standard English?

Reviewer #1: Yes

Reviewer #2: Yes

5. Review Comments to the Author

Reviewer #1: The manuscript fills a significant gap in our understanding of S. aureus by isolating the effects of the agr system from its genetic background. Using congenic Newman strains, the authors demonstrate how different agr-types dictate the timing and intensity of transcriptional responses. The RNA-seq analysis across multiple growth stages offers valuable insights into the divergent regulation of virulence and metabolism.

The manuscript is well written and presents a robust experimental design. I present below a few minor comments that might improve the manuscript:

- In the 12h stationary phase, the authors note different enrichment in arginine biosynthesis. Given the importance of the arginine deiminase system (ADS) in pH homeostasis and survival, a more detailed analysis of the arc operon would strengthen the metabolism aspect of the title.

- The discussion regarding the ecological success of specific lineages remains somewhat speculative. While the congenic approach is a strength for isolating agr function, the "success" of a lineage in a clinical or environmental niche typically results from the co-adaptation of the agr type with its specific genetic background. The authors should explicitly discuss the limitations of using a single background (Newman) to draw broad ecological conclusions.

Reviewer #2: The present manuscript describes an impressive work on the role of the different agr-types over the expression of virulence factors and the overall modification of the transcriptome at three time points (1h30, 6h, and 12h).

The earlier and stronger activation in agr-IV has been previously described, but the authors eliminate the genetic background of the clones presenting the different agr-types to better study the transcriptomic differences associated to the sole variation on the agr system.

The text is clear, figures are easily understandable, and tables offer detailed information.

This reviewer only has a few comments to complete the manuscript:

1. Robustness: The data presented shows differences between agr-IV, and the rest of agr-types, particularly opposite to agr-III.

But all experiments have been performed in triple replicates and to better support the observations more biological replicates seem necessary, particularly for what is shown in Figure 2.

While the differences between agr-types are clear in the relative activation in Figure 1, the described three clusters are not so obvious in Figure 2.

Should the replicates not be feasible at this point, the authors should stress this weakness in the Discussion.

2. In-depth comparison: The authors describe well all the factors that are dramatically highly transcribed in agr-IV isolates when compared to the other agr-types. But further discussion, if any, is missing for the opposite behaviour regarding important S. aureus infection virulence factors such as protein A (spa -surprisingly lower in agr-IV than in any other agr-type), fibronectin-adhering protein (fnbA), chemotaxis-inhibiting protein (chp), which, like the cited superantigens (ssl group) are less expressed in agr-IV than in agr-II or agr-III.

A comment on the putative virulence role of these differences is missing.

3. Clinical significance: It could be interesting to address the probable role of the different transcriptome for the different agr-types in the pathogenesis in humans, for which the following citation is recommended, where early activation of agr-IV pathways is already described, and the importance of agr-III in menstrual toxic shock syndrome is highlighted.

Jarraud S, Lyon GJ, Figueiredo AM, Lina G, Vandenesch F, Etienne J, Muir TW, Novick RP. Exfoliatin-producing strains define a fourth agr specificity group in Staphylococcus aureus. J Bacteriol. 2000 Nov;182(22):6517-22. doi: 10.1128/JB.182.22.6517-6522.2000. Erratum in: J Bacteriol. 2011 Dec;193(24):7027. Gérard, L [corrected to Lina, G]. PMID: 11053400; PMCID: PMC94802.

6. PLOS authors have the option to publish the peer review history of their article (what does this mean?). If published, this will include your full peer review and any attached files.

Reviewer #1: No

Reviewer #2: No

---

## [Author Response · Author response to Decision Letter 1]

24 Apr 2026

We thank the Editor de Lencastre and the Reviewers for their careful reading of our manuscript and for their insightful comments and suggestions. We have revised the manuscript accordingly and believe that these changes have significantly improved the quality and clarity of the work. Below, we provide a detailed, point‑by‑point response to each comment. All changes in the revised manuscript are highlighted in red in the marked manuscript. Then, we mention additional modifications made in the manuscript to follow PLOS One authors guidelines.

Reviewer #1: The manuscript fills a significant gap in our understanding of S. aureus by isolating the effects of the agr system from its genetic background. Using congenic Newman strains, the authors demonstrate how different agr-types dictate the timing and intensity of transcriptional responses. The RNA-seq analysis across multiple growth stages offers valuable insights into the divergent regulation of virulence and metabolism.

The manuscript is well written and presents a robust experimental design. I present below a few minor comments that might improve the manuscript:

- In the 12h stationary phase, the authors note different enrichment in arginine biosynthesis. Given the importance of the arginine deiminase system (ADS) in pH homeostasis and survival, a more detailed analysis of the arc operon would strengthen the metabolism aspect of the title.

We thank Reviewer 1 for highlighting the importance of the ADS, and suggesting to include deeper discussion of its implication in our study. We now discuss a possible acidification of the medium and a switch in the carbon source used by the bacteria (line 211 to 215). In addition to the arc operon, the kdp and ure operons were also upregulated in agr-IV at 12h, although with reduced strength.

- The discussion regarding the ecological success of specific lineages remains somewhat speculative. While the congenic approach is a strength for isolating agr function, the "success" of a lineage in a clinical or environmental niche typically results from the co-adaptation of the agr type with its specific genetic background. The authors should explicitly discuss the limitations of using a single background (Newman) to draw broad ecological conclusions.

We thank Reviewer 1 of highlighting the co-adaptation of the agr-type with genetic background as a missing part in our discussion. We agree with Reviewer 1 and we attenuated the hypothesis of ecological success of given agr-lineages (line 216 to 219) and we explicitly mention the limitation of this study utilizing only one genetic background.

Reviewer #2: The present manuscript describes an impressive work on the role of the different agr-types over the expression of virulence factors and the overall modification of the transcriptome at three time points (1h30, 6h, and 12h).

The earlier and stronger activation in agr-IV has been previously described, but the authors eliminate the genetic background of the clones presenting the different agr-types to better study the transcriptomic differences associated to the sole variation on the agr system.

The text is clear, figures are easily understandable, and tables offer detailed information.

This reviewer only has a few comments to complete the manuscript:

1. Robustness: The data presented shows differences between agr-IV, and the rest of agr-types, particularly opposite to agr-III. But all experiments have been performed in triple replicates and to better support the observations more biological replicates seem necessary, particularly for what is shown in Figure 2.

While the differences between agr-types are clear in the relative activation in Figure 1, the described three clusters are not so obvious in Figure 2.

Should the replicates not be feasible at this point, the authors should stress this weakness in the Discussion.

We thank the Reviewer 2 for raising this concern. Although more replicates would significantly increase the robustness of our results, we believe that the conclusions observed would remain unchanged, considering how biological replicates cluster together on the PC1 axis (Fig 2) and as observed in the new Supplementary Figure S2.

S2 Figure: Cluster identification from PCA analysis

Dot-plots of the principal component analysis (PCA), using PC1 axis, depending on the agr-type, using the entire transcriptome beside the agr-locus, with all time points in A, at time point 1h30 in B, 6h in C and 12h in D. Standard error of the mean (SEM) for each time point in A, and per strain in B, C and D, are represented in black.

Additionally, we have now emphasized how most clusters are observed focusing at the PC1 axis (Fig 2 and S2 Fig), which explains 56 and 51% of the variance at time points 6h and 12h, respectively, compared to only 18 and 10% for the PC2 axis (lines 110 to 112 in the Result section). In addition, we highlight now the variability observed within the clusters, line 115. We also discuss now in the Discussion section the limitation of performing only 3 replicates in this study, lines 216 to 219.

2. In-depth comparison: The authors describe well all the factors that are dramatically highly transcribed in agr-IV isolates when compared to the other agr-types. But further discussion, if any, is missing for the opposite behaviour regarding important S. aureus infection virulence factors such as protein A (spa -surprisingly lower in agr-IV than in any other agr-type), fibronectin-adhering protein (fnbA), chemotaxis-inhibiting protein (chp), which, like the cited superantigens (ssl group) are less expressed in agr-IV than in agr-II or agr-III. A comment on the putative virulence role of these differences is missing.

We appreciate the comment raised by Reviewer 2 regarding specific virulence factors. We added more detailed interpretation of our results in the Discussion section. Notably, we suggest that the downregulation of spa and fnbA in agr-IV is due to a faster activation of the agr-system, which negatively regulates these adhesion factors, and how it can contribute to agr-III virulence, lines 196 to 198. We also note the unexpected observation regarding clfA, lines 201 to 203. Finally, we discussed in more details the observation regarding chp in agr-IV at 6h, line 203 to 206.

3. Clinical significance: It could be interesting to address the probable role of the different transcriptome for the different agr-types in the pathogenesis in humans, for which the following citation is recommended, where early activation of agr-IV pathways is already described, and the importance of agr-III in menstrual toxic shock syndrome is highlighted.

Jarraud S, Lyon GJ, Figueiredo AM, Lina G, Vandenesch F, Etienne J, Muir TW, Novick RP. Exfoliatin-producing strains define a fourth agr specificity group in Staphylococcus aureus. J Bacteriol. 2000 Nov;182(22):6517-22. doi: 10.1128/JB.182.22.6517-6522.2000. Erratum in: J Bacteriol. 2011 Dec;193(24):7027. Gérard, L [corrected to Lina, G]. PMID: 11053400; PMCID: PMC94802.

We thank Reviewer 2 for the suggestion and the reference. We added in the Discussion section that the early activation of agr-IV was previously observed, lines 207 to 208. Regarding the link between agr-types and S. aureus pathogenesis, we mention it in the Introduction, line 57: “Previous studies reported associations between agr-type and infection severity [7–9].”. We now discussed it further in the Discussion section, line 221 to 225.

We would like to bring to the Reviewers and Editor’s attentions that we have added 11 references to support newly discussed points in the Discussion section.

We also added in the Materials and Methods a new subsection: “Artificial Intelligence Tools and Technologies”, detailing the use of AI for this manuscript.

We also updated the R script used for the statistical analysis, which includes the part added to generate graphs for the S2 Figure, and now mentions it in the subsection “RNAseq and Statistical Analysis” of the Materials and Methods section, instead of the Data Availability Statement section.

---

## [Decision Letter · Decision Letter 1]

10 May 2026

*Staphylococcus aureus  agr*-type vs genetic background: molecular signatures determining differential metabolism and virulence potential

PONE-D-26-11159R1

Dear Dr. Mariane Pivard, PhD,

We’re pleased to inform you that your manuscript has been judged scientifically suitable for publication and will be formally accepted for publication once it meets all outstanding technical requirements.

Kind regards,

Herminia de Lencastre, Ph.D.

Academic Editor

PLOS One

Additional Editor Comments (optional):

Reviewers' comments:

Reviewer's Responses to Questions

**Comments to the Author**

1. If the authors have adequately addressed your comments raised in a previous round of review and you feel that this manuscript is now acceptable for publication, you may indicate that here to bypass the “Comments to the Author” section, enter your conflict of interest statement in the “Confidential to Editor” section, and submit your "Accept" recommendation.

Reviewer #1: (No Response)

Reviewer #2: All comments have been addressed

2. Is the manuscript technically sound, and do the data support the conclusions?

Reviewer #1: (No Response)

Reviewer #2: Yes

3. Has the statistical analysis been performed appropriately and rigorously? 

Reviewer #1: (No Response)

Reviewer #2: Yes

4. Have the authors made all data underlying the findings in their manuscript fully available?

Reviewer #1: (No Response)

Reviewer #2: Yes

5. Is the manuscript presented in an intelligible fashion and written in standard English?

Reviewer #1: (No Response)

Reviewer #2: Yes

6. Review Comments to the Author

Reviewer #1: The authors have adequately addressed all previously raised concerns, and the manuscript has improved accordingly. It is now suitable for publication.

Reviewer #2: (No Response)

7. PLOS authors have the option to publish the peer review history of their article (what does this mean?). If published, this will include your full peer review and any attached files.

Reviewer #1: No

Reviewer #2: No

---

## [Editor Report · Acceptance letter]

PONE-D-26-11159R1

PLOS One

Dear Dr. Pivard,

I'm pleased to inform you that your manuscript has been deemed suitable for publication in PLOS One. Congratulations! Your manuscript is now being handed over to our production team.

Kind regards,

on behalf of

Dr. Herminia de Lencastre

Academic Editor

PLOS One